# Realization of the Sensorless Permanent Magnet Synchronous Motor Drive Control System with an Intelligent Controller

**Hung-Khong Hoai [1,2], Seng-Chi Chen [1,\*] and Hoang Than [3]**

[1] Department of Electrical Engineering, Southern Taiwan University of Science and Technology, Tainan 71005, Taiwan; da62b201@stust.edu.tw
[2] Faculty of Electrical and Electronics Engineering, Ho Chi Minh City University of Transport, Ho Chi Minh City 70000, Vietnam
[3] Department of Electrical and Electronics Engineering, Hue Industrial College, Hue 49000, Vietnam; hthan@hueic.edu.vn
\* Correspondence: amtfcsg123@stust.edu.tw; Tel.: +886-6-253-3131 (ext. 3324)

**Abstract:** This paper presents the sensorless control algorithm for a permanent magnet synchronous motor (PMSM) drive system with the estimator and the intelligent controller. The estimator is constructed on the novel sliding mode observer (SMO) in combination with a phase-locked loop (PLL) to estimate the position and speed of the rotor. The intelligent controller is a radial basis function neural network (RBFNN)-based self-tuning PID (Proportional-Integral-Derivative) controller, applied to the velocity control loop of the PMSM drive control system to adapt strongly to dynamic characteristics during the operation with an external load. The *I-f* startup strategy is adopted to accelerate the motor from standstill, then switches to the sensorless mode smoothly. The control algorithm program is based on MATLAB and can be executed in simulations and experiments. The control system performance is verified on an experimental platform with various speeds and the dynamic load, in which the specified *I-f* startup mode and sensorless mode, inspected by tracking response and speed regulation. The simulation and experimental results demonstrate that the proposed method has worked successfully. The motor control system has smooth switching, good tracking response, and robustness against disturbance.

**Keywords:** permanent magnet synchronous motor; sensorless control; sliding mode observer; RBFNN-based self-tuning PID controller; *I-f* startup strategy

## 1. Introduction

Applying the advanced developments in power electronics, microprocessors and digital signal processors (DSP), permanent magnet synchronous motors (PMSMs) have been extensively used in industrial automatic control applications, from washing machines to electric vehicles, due to their efficient performance characteristics, high power transmission efficiency, large torque-to-weight ratio, and long service life. Among various PMSM drive control techniques, field-oriented control (FOC) has been the most essential and efficient scheme, in which the rotor's position and speed data are required. As a sensor-based solution, these data are managed by a typical sensor, such as a resolver, encoder, or Hall sensor, installed on the motor's shaft. However, this makes the PMSM drive control system more expensive and larger in size. In some cases, the sensors are environment-sensitive, reducing control reliability and adaptiveness. To solve these shortcomings, various back-EMF-based (back electromotive force based) sensorless control techniques are designed and applied, such as the extended Kalman filter (EKF) approaches [1–3] or the sliding mode observer (SMO) approaches [4–7]. The EKF involves

a lot of recursive computations because it consists of prediction and innovation. Meanwhile, the SMO is robust against disturbance, has a high accuracy estimation potential, and is easy to be implemented. Generally, the conventional SMO solutions are impacted by chattering problems and noise effects because of using the on-off function and traditional arctangent calculation, so that the novel SMO and PLL combinations are widely applied to enhance the estimator's robustness and accuracy [6,8–10]. Due to the back EMF value being small at the standstill or the low-speed region, the startup strategies are implemented to speed up the motor to the specific speed threshold at which the back-EMF is large enough to be estimated precisely. A simple *I-f* startup strategy was presented [11] and applied [1,5]. It involves the closed-loop current control to ensure the motor starts successfully under different external load situations without initial rotor position estimation and machine parameters estimation.

A proper motor drive control system requires a wide adjustable speed range, adaption to load disturbances and parameter variations, high instantaneous torque response, and lower torque ripple during the operating condition. It is essential to improve the motor control algorithm to obtain optimal performance. Therefore, various controllers have been proposed, such as the ANFIS controller [12], the neural network controller [13], the hybrid fuzzy-PID (Proportional-Integral-Derivative) controller [14], and the backstepping controller [4]. Among these algorithms, the radial basis function neural network (RBFNN)-based self-tuning PID controller is considered to enhance the speed control quality of the PMSMs drive control system. This intelligent controller not only inherits the typical PID controller's structural simplicity but also is optimized by online adjusting the operating parameters based on the advantages of a neural network such as the ability to identify nonlinear system dynamics, the ability to learn, generalize, and adapt.

In the realization of the motor control system, the DSP of Texas Instruments provides a flexible solution, improving system reliability and efficiency. The DSP integrates highly optimized peripheral circuits, memory, and a single-chip CPU structure. It exhibits powerful processing and high performance for complex real-time control systems. In particular, with MATLAB's Embedded Coder Support Package utilities, it is resourceful to build up the motor control algorithm in Simulink. The DSP application development time is significantly shortened. Accordingly, in this paper, an RBFNN-based self-tuning PID speed controller is implemented to improve the performance of the DSP-based sensorless PMSM drive control system with an estimator based on the novel SMO in combination with a PLL (the novel SMO-PLL). The MATLAB-based implemented control algorithm is not only executed in simulation but also applied to the real-time experiment system. The *I-f* startup mode, tracking response, and speed regulation are investigated to evaluate the control system performance. The proposed control algorithm is verified on an experimental platform with a PMSM, inverter, control circuit, a Texas Instruments DSP F28379D, and the dynamic load.

This paper is organized into the following sections. Section 2 describes the sensorless PMSM drive control system with the mathematical model, the estimator based on the novel SMO-PLL, and the *I-f* startup strategy. Section 3 introduces the self-tuning PID controller based on an RBFNN. Section 4 presents the implementation of the control algorithm in MATLAB Simulink. In Section 5, the control system performance is inspected in the digital simulation to evaluate the correctness and the effectiveness of the control algorithm. In Section 6, the proposed algorithm is verified on the experimental platform. Finally, the conclusion is given in Section 7.

## 2. Description of the Sensorless PMSM Drive System

The architecture of the proposed DSP-based sensorless PMSM drive control system based on the novel SMO-PLL estimator and the RBFNN-based self-tuning PID controller is shown in Figure 1. As mentioned above, two control modes are investigated from low speed to high speed, including a startup mode and a sensorless mode. The first mode is the *I-f* startup strategy mode in the low-speed range. The switches are activated in position 1. The second mode is the sensorless control mode in the high-speed range. The switches are activated in position 2. In the sensorless control mode, there are two closed control loops—the inner current control loop and the outer velocity control

loop. The current control loops are executed on the PI (Proportional-Integral) controller and the FOC algorithm. The velocity control loop is implemented by an RBFNN-based self-tuning PID controller. The detailed formulation and control algorithm are described as follows.

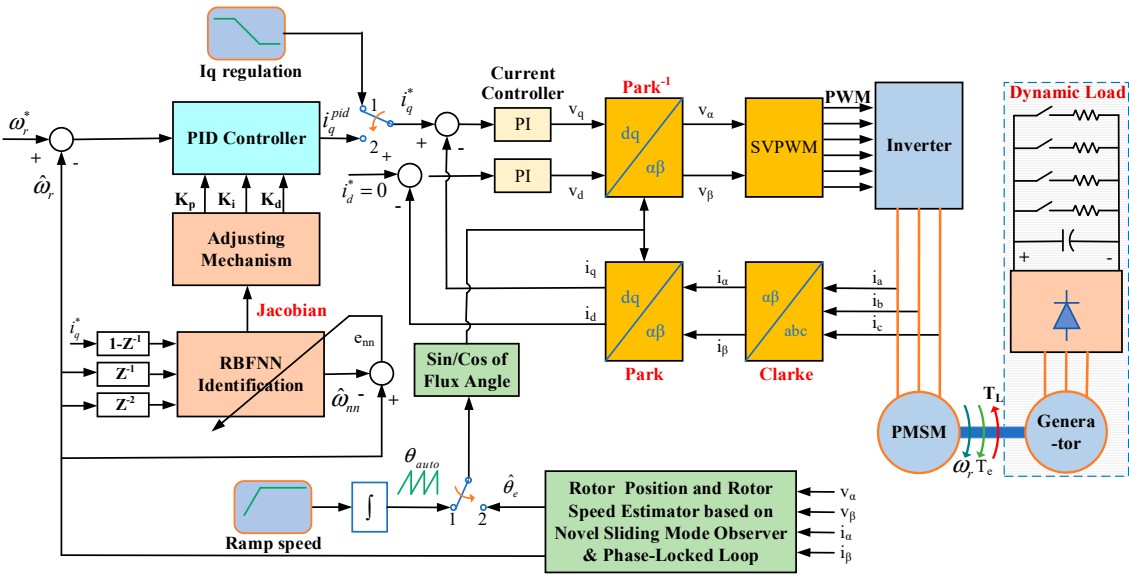

**Figure 1.** The architecture of a self-tuning PID (Proportional-Integral-Derivative) speed controller based on the radial basis function neural network (RBFNN) for the sensorless permanent magnet synchronous motor (PMSM) drive control system with the dynamic load.

### 2.1. Modeling of the PMSM

In general, the typical mathematical model of a PMSM is formulated in the *d-q* synchronous rotating coordinate as:

$$\begin{cases} v_d = r_s i_d + L_s \frac{d}{dt} i_d - \omega_e L_s i_q \\ v_q = r_s i_q + L_s \frac{d}{dt} i_q + \omega_e L_s i_d + \omega_e \lambda_f \end{cases} \tag{1}$$

where $v_d$, $v_q$ are the voltages of the *d* and *q* axis, respectively; $r_s$ is the stator winding resistance per phase; $L_s = L_d = L_q$ (for SPMSM) and $L_d$, $L_q$ are the inductances of the *d* and *q*, axis respectively; $i_d$, $i_q$ are the currents of the *d* and *q* axis respectively; $\omega_e$ is the rotating speed of magnet flux; $\lambda_f$ is the permanent magnet flux linkage.

In Figure 1, the current control of PMSM is decoupled to torque control ($i_q$) and flux control ($i_d$) based on the Clarke and Park transformation. Therefore, the PMSM is controlled in a similar way to controlling a DC motor. When the current $i_d$ is controlled to zero, the torque of PMSM can be expressed by the following equation:

$$T_e = \frac{3N_p}{4} \lambda_f i_q \triangleq K_t i_q \tag{2}$$

with:

$$K_t = \frac{3N_p}{4} \lambda_f \tag{3}$$

where $T_e$ is the electromagnetic torque, $K_t$ is the torque constant, $N_p$ is the pole pairs.

The dynamic equation of a PMSM drive system with the mechanical load can be presented as:

$$\frac{d\omega_r}{dt} = \frac{1}{J}(T_e - T_L - B\omega_r) \tag{4}$$

where $\omega_r$ is the rotor speed, $T_L$ is the load torque; $J$ and $B$ are the inertia and friction constants of the PMSM, respectively.

*2.2. Novel Sliding Mode Observer*

Applying the inverse Park transformation for (1), the circuit equation of PMSM on the $\alpha - \beta$ stationary coordinate can be represented by the following equation:

$$\begin{cases} v_\alpha = r_s i_\alpha + L_s \frac{d}{dt} i_\alpha + e_\alpha \\ v_\beta = r_s i_\beta + L_s \frac{d}{dt} i_\beta + e_\beta \end{cases} \tag{5}$$

where $i_\alpha, i_\beta$ are the current, $v_\alpha, v_\beta$ are the voltage of the $\alpha$ and $\beta$ axis, and $e_\alpha, e_\beta$ are the back EMF, which are expressed by

$$\begin{cases} e_\alpha = -\omega_e \lambda_f \sin \theta_e \\ e_\beta = \omega_e \lambda_f \cos \theta_e \end{cases} \tag{6}$$

where $\theta_e$ is the electrical rotor position and $\dot{\theta}_e = \omega_e$.

Combining (5) and (6), the current equation can be derived as:

$$\begin{cases} \frac{d}{dt} i_\alpha = \frac{1}{L_s}(-r_s i_\alpha + v_\alpha - e_\alpha) \\ \frac{d}{dt} i_\beta = \frac{1}{L_s}(-r_s i_\beta + v_\beta - e_\beta) \end{cases} \tag{7}$$

According to the novel sliding mode observer theory, the current observer formulation is written as follows:

$$\begin{cases} \frac{d}{dt} \hat{i}_\alpha = \frac{1}{L_s}(-r_s i_\alpha + v_\alpha - kH(\hat{i}_\alpha - i_\alpha)) \\ \frac{d}{dt} \hat{i}_\beta = \frac{1}{L_s}(-r_s i_\beta + v_\beta - kH(\hat{i}_\beta - i_\beta)) \end{cases} \tag{8}$$

where $\hat{i}_\alpha, \hat{i}_\beta$ are the estimated current in the $\alpha$ and $\beta$ axis. The $k$ is the gain and $H(x)$ is the Sigmoid function, which is defined as:

$$H(x) = \frac{2}{1 + e^{-2\mu x}} - 1 = \frac{1 - e^{-2\mu x}}{1 + e^{-2\mu x}} \tag{9}$$

where $\mu$ is the shape parameter. Moreover, when (8) subtracts (7), the estimated current error can be represented in the dynamic equation as the following equation:

$$\begin{cases} \frac{d}{dt} \widetilde{i}_\alpha = \frac{1}{L_s}(-r_s \widetilde{i}_\alpha + e_\alpha - kH(\widetilde{i}_\alpha)) \\ \frac{d}{dt} \widetilde{i}_\beta = \frac{1}{L_s}(-r_s \widetilde{i}_\beta + e_\beta - kH(\widetilde{i}_\beta)) \end{cases} \tag{10}$$

where $\widetilde{i}_\alpha, \widetilde{i}_\beta$ are the estimated current error in the $\alpha$ and $\beta$ axis, and are defined as: $\widetilde{i}_\alpha = \hat{i}_\alpha - i_\alpha$, and $\widetilde{i}_\beta = \hat{i}_\beta - i_\beta$.

To analyze the stability of the novel sliding mode observer, the Lyapunov function is chosen as:

$$V = \frac{1}{2}(\widetilde{i}_\alpha^2 + \widetilde{i}_\beta^2) \tag{11}$$

Taking the derivative of the Lyapunov function by the time, we obtain

$$\dot{V} = \widetilde{i}_\alpha \times \frac{d}{dt} \widetilde{i}_\alpha + \widetilde{i}_\beta \times \frac{d}{dt} \widetilde{i}_\beta \tag{12}$$

Furthermore, if we design $k > max(|e_\alpha|, |e_\beta|)$ and use (10), we have

$$\begin{cases} \widetilde{i}_\alpha \times \frac{d}{dt} \widetilde{i}_\alpha = \frac{\widetilde{i}_\alpha}{L_s}(-r_s \widetilde{i}_\alpha + e_\alpha - kH(\widetilde{i}_\alpha)) \leq -\frac{r_s}{L_s} \widetilde{i}_\alpha^2 \leq 0 \\ \widetilde{i}_\beta \times \frac{d}{dt} \widetilde{i}_\beta = \frac{\widetilde{i}_\beta}{L_s}(-r_s \widetilde{i}_\beta + e_\beta - kH(\widetilde{i}_\beta)) \leq -\frac{r_s}{L_s} \widetilde{i}_\beta^2 \leq 0 \end{cases} \tag{13}$$

Therefore, $\dot{V} \leq 0$, the estimated current will converge to the actual current. The back EMF can be obtained as:

$$\begin{cases} e_\alpha \approx \hat{e}_\alpha = kH(\hat{i}_\alpha - i_\alpha) = kH(\widetilde{i}_\alpha) \\ e_\beta \approx \hat{e}_\beta = kH(\hat{i}_\beta - i_\beta) = kH(\widetilde{i}_\beta) \end{cases} \tag{14}$$

where $\hat{e}_\alpha, \hat{e}_\beta$ are the estimated back EMF.

Moreover, a low pass filter is implemented to reduce the noise effect in estimated back EMF.

$$\begin{cases} \hat{e}_{\alpha F} = \frac{\omega_c}{s + \omega_c} \hat{e}_\alpha \\ \hat{e}_{\beta F} = \frac{\omega_c}{s + \omega_c} \hat{e}_\beta \end{cases} \tag{15}$$

where $\omega_c = 2\pi f_c$ and $f_c$ is the cut-off frequency of the low pass filter.

Once the back EMF can be estimated, the phase-locked loop is executed to estimate the rotor position. The architecture of the novel SMO with a PLL is shown in Figure 2.

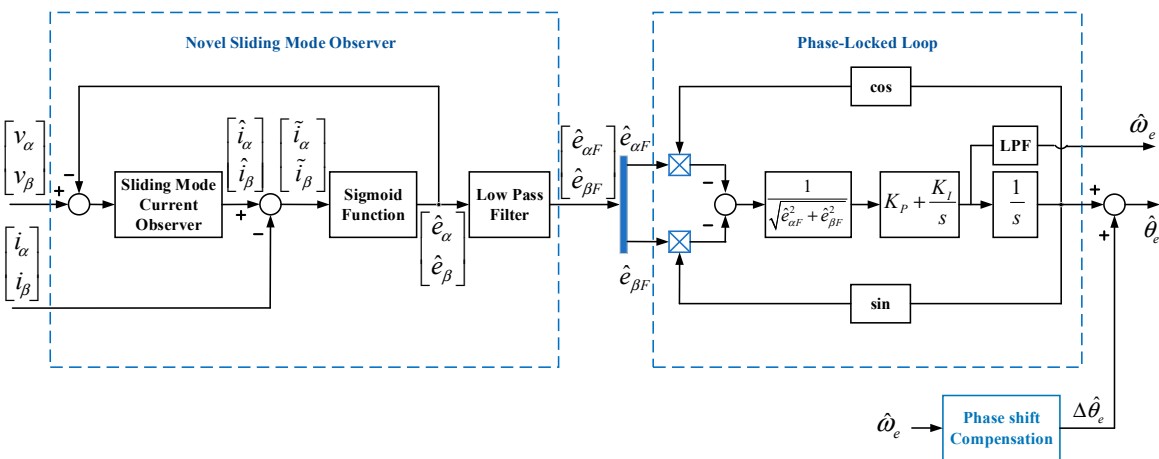

**Figure 2.** Novel sliding mode observer with a phase-locked loop.

From (6), an estimated error is defined by

$$\varepsilon = -\hat{e}_{\alpha F} cos(\hat{\theta}_e) - \hat{e}_{\beta F} sin(\hat{\theta}_e) \tag{16}$$

If this error is controlled to zero by a PI controller, the rotor speed is estimated by

$$\begin{cases} \hat{\omega}_e = K_P f_I(\varepsilon) + K_i \int f_I(\varepsilon) dt \\ f_I(\varepsilon) = \frac{\varepsilon}{\sqrt{\hat{e}_{\alpha F}^2 + \hat{e}_{\beta F}^2}} \end{cases} \tag{17}$$

and, the estimated rotor position can be obtained by

$$\hat{\theta}_e = \int \hat{\omega}_e dt \tag{18}$$

Finally, a phase shift compensation is calculated to correct the estimated rotor position, rejecting the phase delay effect in the back EMF's low pass filter.

$$\Delta \hat{\theta}_e = arctan\left(\frac{\hat{\omega}_e}{\omega_c}\right) \tag{19}$$

### 2.3. The I-f Startup Strategy

During the inter-mode transition from the startup stage to the sensorless FOC algorithm stage, the *I-f* startup strategy smooths torque and speed transition. It is based on the closed-loop current controller architecture. According to the study of [11], the *I-f* startup method is implemented in a three-stepped sequence, shown in Figure 3, and described as follows:

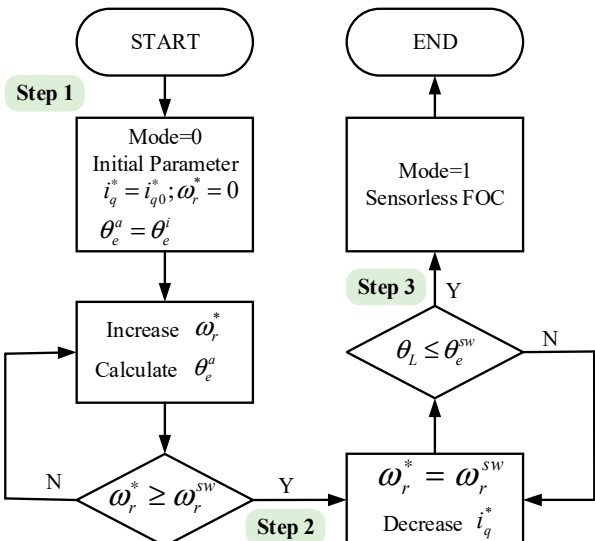

**Figure 3.** The *I-f* startup strategy flowchart.

**Step 1:** Ramping up reference speed with the constant $i_q^*$ regulation at the initial start. This step aims to accelerate the motor from standstill to the switching reference speed where the back-EMF signal is large enough for the novel SMO-PLL estimator, which can obtain the rotor position and speed information accurately. The reference current $i_q^*$ and reference speed $\omega_r^*$ are set to:

$$i_q^*(k) = i_{q0}^* \tag{20}$$

$$\omega_r^*(k+1) = \begin{cases} \omega_r^*(k) + K_{ramp}T_s & if\ \omega_r^* < \omega_r^{sw} \\ \omega_r^{sw} & if\ \omega_r^* \geq \omega_r^{sw} \end{cases} \tag{21}$$

$$\theta_e^a(k+1) = \theta_e^a(k) + \omega_r^*T_s \quad with\ \theta_e^a(0) = \theta_e^i \tag{22}$$

where $i_{q0}^*$ is the initial current in q-axis; $\theta_e^a, \theta_e^i$ are the auto-generated rotor position and its initial value; $\omega_r^{sw}$ is the specific reference speed for switching to the sensorless mode, and $K_{ramp}$ is the proportional gain for ramping up the reference rotor speed.

**Step 2:** Keeping reference speed constant with decreasing $i_q^*$ axis current at the end of step 1. The purpose of this step is to reduce the current $i_q^*$, while the same torque is generated as before and balanced the load torque. In addition, the load torque angular $\theta_L$ reduces to a critical threshold to guarantee the smooth switching condition.

$$i_q^*(k+1) = i_q^*(k) - K_aT_s \tag{23}$$

$$\omega_r^*(k+1) = \omega_r^*(k) = \omega_r^{sw} \tag{24}$$

while:

$$\theta_L = \hat{\theta}_e - \theta_e^a > \theta_e^{sw} \tag{25}$$

where $\theta_e^{sw}$ is the angular threshold value of switching condition and $K_a$ is the proportional gain for ramping down the current command $i_q^*$.

**Step 3:** Switching to the sensorless FOC mode operation based on the novel SMO-PLL estimator at the end of step 2.

## 3. The Self-Tuning PID Controller Based on an RBFNN

### 3.1. PID Controller

In Figure 1, the velocity loop is executed with a PID controller to regulate the reference current $i_q^*$ in the $q$-axis. The algorithm of an incremental PID controller can be written as the following discrete-time equation:

$$i_q^*(k) = i_q^*(k-1) + k_p \widetilde{e_p}(k) + k_i \widetilde{e_i}(k) + k_d \widetilde{e_d}(k) \tag{26}$$

with:

$$\begin{cases} \widetilde{e_p}(k) = e(k) - e(k-1) \\ \widetilde{e_i}(k) = e(k) \\ \widetilde{e_d}(k) = e(k) - 2e(k-1) + e(k-2) \end{cases} \tag{27}$$

where the rotor speed error is defined as:

$$e(k) = \omega_r^*(k) - \hat{\omega}_r(k) \tag{28}$$

and $k_p$, $k_i$, $k_d$ are the proportional, integral, and differential gains of the controller, respectively. Meanwhile $\widetilde{e_p}(k), \widetilde{e_i}(k), \widetilde{e_d}(k)$ are the proportional, integral, and differential variations in one sampling time, successively.

### 3.2. Radial Basis Function Neural Network

To get a self-tuning PID controller, the PMSM drive control system is identified by an RBFNN. Then, the controller's parameters should be adjusted suitably. Therefore, the RBFNN structure is shown in Figure 4. It comprises the feedforward neural network architecture, consisting of three layers—an input layer with three inputs, a hidden layer with five neurons, and a single output layer.

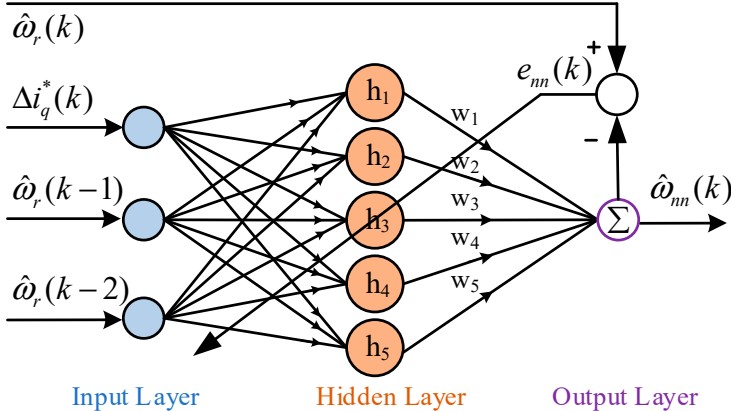

**Figure 4.** The RBFNN for tuning the PID parameters.

The input vector is defined as:

$$x(k) = \left[ \Delta i_q^*(k) \;\; \hat{\omega}_r(k-1) \;\; \hat{\omega}_r(k-2) \right]^T \tag{29}$$

with:

$$\Delta i_q^*(k) = i_q^*(k) - i_q^*(k-1) \tag{30}$$

In the hidden layer, the activation function is implemented by a Gaussian function

$$h_j(k) = exp\left(-\frac{x(k) - C_j(k)^2}{2b_j^2(k)}\right) \tag{31}$$

where $C_j(k) = \begin{bmatrix} c_{1j}(k) & c_{2j}(k) & c_{3j}(k) \end{bmatrix}^T$ and $b_j$, $c_{ij}$ are the width and center of the Gaussian function.

In the output layer, the output is a linear sum of hidden nodes:

$$\hat{\omega}_{nn}(k) = \sum_{j=1}^{5} w_j(k)h_j(k) \tag{32}$$

Furthermore, to simplify the online adaptive updating law for the parameter of RBFNN, the cost function is defined as:

$$J(k) = \frac{1}{2}(\hat{\omega}_r(k) - \hat{\omega}_{nn}(k))^2 = \frac{1}{2}e_{nn}^2(k) \tag{33}$$

According to the stochastic gradient descent (SGD) method, the learning algorithm can be calculated and updated by the below equations [15]:

$$\begin{cases} \Delta\omega_j(k) = -\eta\frac{\partial J(k)}{\partial \omega_j(k)} = \eta e_{nn}(k)\omega_j(k) \\ \omega_j(k) = \omega_j(k-1) + \Delta\omega_j(k) + \alpha(\omega_j(k-1) - \omega_j(k-2)) \end{cases} \tag{34}$$

$$\begin{cases} \Delta b_j(k) = -\eta\frac{\partial J(k)}{\partial b_j(k)} = \eta e_{nn}(k)\omega_j(k)h_j(k)\frac{x(k) - C_j(k)^2}{2b_j^3(k)} \\ b_j(k) = b_j(k-1) + \Delta b_j(k) + \alpha(b_j(k-1) - b_j(k-2)) \end{cases} \tag{35}$$

$$\begin{cases} \Delta c_{ij}(k) = -\eta\frac{\partial J(k)}{\partial c_{ij}(k)} = \eta e_{nn}(k)\omega_j(k)\frac{x_i(k) - c_{ij}(k)}{b_j^2(k)} \\ c_{ij}(k) = c_{ij}(k-1) + \Delta c_{ij}(k) + \alpha(c_{ij}(k-1) - c_{ij}(k-2)) \end{cases} \tag{36}$$

where $i = 1, 2, 3$; $j = 1, 2...5$; $\eta$ is the learning rate; $\alpha$ is the momentum factor.

Furthermore, the Jacobian transformation can be obtained as:

$$\frac{\partial\hat{\omega}_r}{\partial\Delta i_q^*} \approx \frac{\partial\hat{\omega}_{nn}}{\partial\Delta i_q^*} = \sum_{j=1}^{5} w_j(k)h_j(k)\frac{c_{1j}(k) - \partial\Delta i_q^*(k)}{b_j^2(k)} \tag{37}$$

### 3.3. Adjusting Mechanism Of PID Controller

In the closed-loop control, the parameters of the PID controller are adjusted online to minimize the square rotor speed error between the reference speed and the estimated rotor speed. Therefore, the cost function can be expressed as:

$$E(k) = \frac{1}{2}(\omega_r^*(k) - \hat{\omega}_r(k))^2 = \frac{1}{2}e^2(k) \tag{38}$$

Then, using the SGD method, the parameters of the PID controller can be updated and optimally tuned as the following equations [16]:

$$\begin{cases} \Delta k_p = -\eta\frac{\partial E}{\partial k_p} = \eta\frac{\partial E}{\partial\hat{\omega}_r}\frac{\partial\hat{\omega}_r}{\partial\Delta i_q^*}\frac{\partial\Delta i_q^*}{\partial k_p} \approx \eta e(k)\frac{\partial\hat{\omega}_{nn}}{\partial\Delta i_q^*}\widetilde{e}_p \\ k_p(k) = k_p(k-1) + \Delta k_p \end{cases} \tag{39}$$

$$\begin{cases} \Delta k_i = -\eta\frac{\partial E}{\partial k_i} = \eta\frac{\partial E}{\partial\hat{\omega}_r}\frac{\partial\hat{\omega}_r}{\partial\Delta i_q^*}\frac{\partial\Delta i_q^*}{\partial k_i} \approx \eta e(k)\frac{\partial\hat{\omega}_{nn}}{\partial\Delta i_q^*}\widetilde{e}_i \\ k_i(k) = k_i(k-1) + \Delta k_i \end{cases} \tag{40}$$

$$\begin{cases} \Delta k_d = -\eta \frac{\partial E}{\partial k_d} = \eta \frac{\partial E}{\partial \hat{\omega}_r} \frac{\partial \hat{\omega}_r}{\partial \Delta i_q^*} \frac{\partial \Delta i_q^*}{\partial k_d} \approx \eta e(k) \frac{\partial \hat{\omega}_{nn}}{\partial \Delta i_q^*} \widetilde{e}_d \\ k_d(k) = k_d(k-1) + \Delta k_d \end{cases} \tag{41}$$

where $\eta$ is the learning rate.

## 4. Implementation of a Sensorless Speed Control Algorithm in MATLAB Simulink

Among the motor control system realization approaches, the DSP-executed hardware solution requires transforming the system description equation from the continuous-time domain to the discrete-time domain. The sliding mode observer is represented by the discrete-time equation as:

$$\begin{cases} \hat{i}_\alpha(k+1) = e^{-\frac{r_s}{L_s}T_s}\hat{i}_\alpha(k) + \frac{1}{r_s}\left(1 - e^{-\frac{r_s}{L_s}T_s}\right)(v_\alpha(k) - \hat{e}_\alpha(k)) \\ \hat{i}_\beta(k+1) = e^{-\frac{r_s}{L_s}T_s}\hat{i}_\beta(k) + \frac{1}{r_s}\left(1 - e^{-\frac{r_s}{L_s}T_s}\right)(v_\beta(k) - \hat{e}_\beta(k)) \end{cases} \tag{42}$$

where $T_s$ is the sampling time.

The motor control system expression is standardized in the per-unit system, so the current observer is rewritten as:

$$\begin{cases} \hat{i}_{\alpha,pu}(k+1) = F\hat{i}_{\alpha,pu}(k) + G(v_{\alpha,pu}(k) - \hat{e}_{\alpha,pu}(k)) \\ \hat{i}_{\beta,pu}(k+1) = F\hat{i}_{\beta,pu}(k) + G(v_{\beta,pu}(k) - \hat{e}_{\beta,pu}(k)) \end{cases} \tag{43}$$

with: $F = e^{-\frac{r_s}{L_s}T_s}$ and $G = \frac{U_m}{I_m}\frac{1}{r_s}\left(1 - e^{-\frac{r_s}{L_s}T_s}\right)$

where $U_m$ and $I_m$ are the base phase voltage and current, $F$ and $G$ are the feedback and gain factors of the SMO block system, respectively.

Additionally, the back EMF's low pass filter is also represented by the discrete equation, as:

$$\begin{cases} \hat{e}_{\alpha F}(k+1) = \hat{e}_{\alpha F}(k) + K_{LPF}[\hat{e}_\alpha(k) - \hat{e}_{\alpha F}(k)] \\ \hat{e}_{\beta F}(k+1) = \hat{e}_{\beta F}(k) + K_{LPF}[\hat{e}_\beta(k) - \hat{e}_{\beta F}(k)] \end{cases} \tag{44}$$

where $K_{LPF} = \omega_c T_s$.

Generally, a DSP-based control system is programmed in C/C++ languages, which requires some coding and debugging skills. However, in the case of using the Texas Instruments' DSP controller, the algorithm is not only being implemented directly by Code Composer Studio (CCS) software but is also enhanced by utilizing MATLAB and CCS software. The algorithm is firstly designed in MATLAB Simulink, then compiled the block system into C/C++ languages by Embedded Coder. Successively, the code is imported into CCS to execute, debug, and monitor the real-time system. Moreover, with enhanced functions of the Embedded Coder Support Package for Texas Instruments in MATLAB, it is very convenient to develop a motor control algorithm with Simulink. This shortens the DSP application development time significantly.

For developing the FOC algorithm, the Digital Control Motor (DCM) blocks are provided with the pre-built functions of not only the Clarke, Park, and Inverse Park transformations, but also for SVPWM, and current PI controllers. The FOC block is not shown in this paper with these functions, because it is already described in MATLAB manuals. Moreover, the TI IQmath library is supported for highly optimized and high precision mathematical functions of cosine, sine, magnitude, etc. Therefore, the sensorless PMSM control algorithm is easily developed in MATLAB Simulink for both simulation and experiment. Figure 5 presents the detailed block diagram of the novel SMO, the sigmoid function, the low pass filter, and the PLL in Simulink.

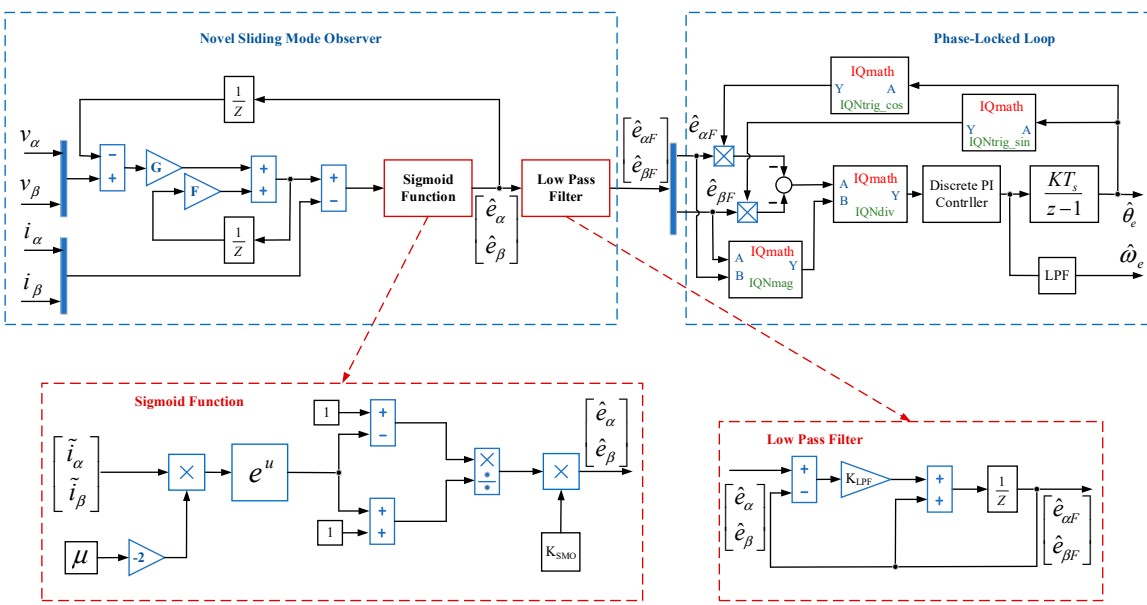

**Figure 5.** The block diagram of the novel sliding mode observer (SMO) with a phase-locked loop (PLL) in Simulink.

According to the design technique in the literature [15,17], the discrete PID controller algorithm in (26)–(28), (39)–(41) and an RBFNN identification algorithm in (29)–(37) are implemented as two function blocks, integrated in the closed velocity control loop (Figure 6).

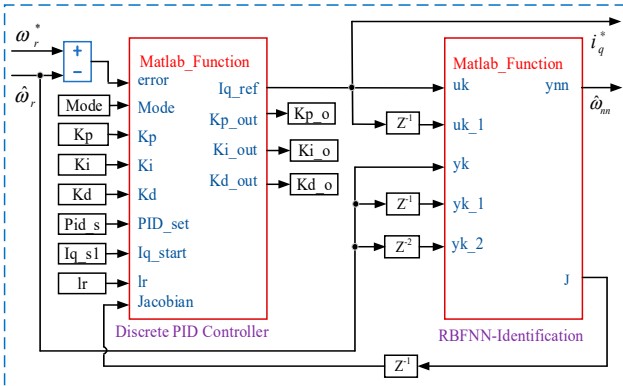

**Figure 6.** The block diagram of PID and RBFNN based on MATLAB function.

## 5. The Simulation Results

The platform of a motor speed control system was built on the PMSM's parameters, shown in Table 1. The control algorithm is verified in both the simulation and experiment under three cases. The first case is the *I-f* startup mode and switching to the sensorless FOC algorithm. The second one is the tracking response for checking the transient specification and the steady-state error. The last one is the speed regulation for evaluating the system stability regarding external disturbances. All the instances are operated with the dynamic load system, which includes an electrical circuit and a generator, coupled to the PMSM. The electrical circuit consists of a rectifier, a capacitor, and resistors, which are connected in parallel.

**Table 1.** Parameters.

| Parameters | Values |
|---|---|
| Rated power | 750 W |
| Input voltage | 220 VAC |
| Rated current | 4.24 A |
| Rated speed | 2000 rpm |
| Rated torque | 3.655 Nm |
| Phase resistance | 1.326 Ω |
| Phase inductance | 2.952 mH |
| BEMF constant | 56.5 ($V_{L-L}$/krpm) |
| Torque constant | 0.86 Nm/A |
| Inertia | 3.63 Kgcm$^2$ |
| Slot/poles | 12S/8P |

The complete design of the sensorless PMSM drive control system is shown in Figure 7, which includes three blocks. The first block (A) is built for real-time platform modeling. The discrete times of the PMSM model, the inverter, and the generator are 50 µs. The second block (B) is the motor control algorithm, which is executed in simulation and the real-time experiment. The third block (C) is used to monitor and acquire the simulation data. Within the control algorithm block, five subblocks are executed as the reference speed regulator, the discrete RBFNN-based self-tuning PID controller, the $i_q^*$ regulator of *I-f* mode, the FOC algorithm, and the novel sliding mode observer with a phase-locked loop, respectively. The sampling time of the velocity loop is 1ms, while the sampling times of the FOC algorithm and the novel SMO-PLL estimator are 50 µs. In the current control loop, two PI controllers are adopted for the currents in *d-q* axis. Those PI controllers' parameters are set as $K_{Pd} = 0.25$, $K_{Id} = 0.025$, $K_{Pq} = 0.25$, $K_{Iq} = 0.025$. In the speed control loop, it is implemented by a self-tuning PID controller based on an RBFNN. The neuron parameters of those compositional five hidden nodes are initially set, as node centers ($c_1 = -0.01$, $c_2 = -0.005$, $c_3 = 0.0$, $c_4 = 0.005$, $c_5 = 0.01$), node widths ($b_1 = b_2 = b_3 = b_4 = b_5 = 0.005$), connective weights ($w_1 = w_2 = w_3 = w_4 = w_5 = 0.000625$). The learning rate is 0.475, and the momentum factor is 0.95. Additionally, the discrete PID's initial parameters are designed as $K_{Ps} = 0.725$, $K_{Is} = 0.0105$, $K_{Ds} = 0.483$. These initial values are tuned during the operating time. The dynamic load system is set up with an initial resistance load of 100 Ω and a capacitor of 470 µF.

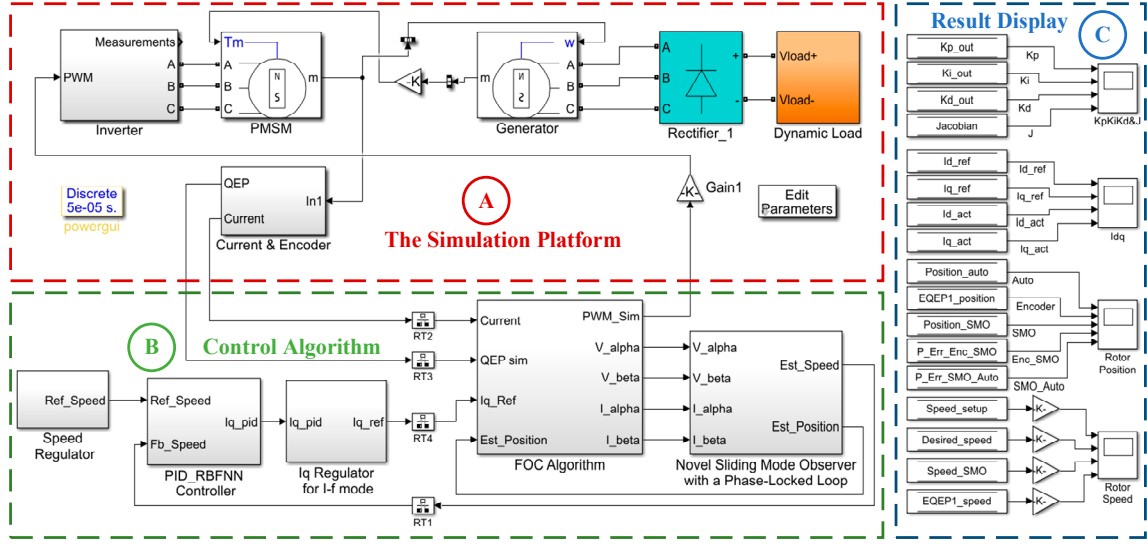

**Figure 7.** The simulation structure of the sensorless speed control algorithm for the PMSM drive system with a dynamic load system in MATLAB Simulink.

Figure 8 shows the motor's speed responses while speeding up the rotor speed from 0 to 2000 rpm. The reference speed is changed in a period of 1s with a sequence of 200 → 400 → 700 → 1000 → 700 → 1000 → 1400 → 1700 → 2000 rpm. Figure 8a refers that the novel SMO-PLL estimated rotor speed is closely tracked to the reference speed, and overlaps the actual rotor speed, calculated by the encoder. Figure 8b presents the current response in the *d-q* axis. The current $i_q$ is varied and regarded to the reference speed. When the rotor speed is increased, the external load becomes higher. Therefore, the more torque is required in the motor, and the current $i_q$ is regulated to be larger. The current $i_d$ almost is equal to zero and has some small pulses when changing the speed. Figure 8c–f illustrates the electrical positions of the rotor, separately calculated by the novel SMO-PLL estimator and the encoder, with the estimated errors at the speeds of 700, 1000, 1400, and 2000 rpm. There are 7, 10, 14 and 20 position cycles in a period of 0.15s, respectively. Correspondingly, the rotation frequencies are 46.67, 66.67, 93.33, and 133.33 Hz. These values are suitable for the mentioned rotor speed. The estimated and actual positions are approximated to each other; thus, the estimated error is equal to zero. The PID gains are updated during the operating time. At = 3.5 s (as $\omega_r$ = 700 rpm), the PID gains are tuned to $K_{Ps}$ = 0.7424, $K_{Is}$ = 0.0107, and $K_{Ds}$ = 0.4806. At = 4.5 s (as $\omega_r$ = 1000 rpm), the PID gains are tuned to $K_{Ps}$ = 0.7694, $K_{Is}$ = 0.0110, and $K_{Ds}$ = 0.4768. At t = 7.5 s (as $\omega_r$ = 1400 rpm), the PID gains are tuned to $K_{Ps}$ = 0.8549, $K_{Is}$ = 0.0117, and $K_{Ds}$ = 0.4648. Lastly, the rotor speed reaches 2000 rpm and the PID gains are also tuned to $K_{Ps}$ = 1.0288, $K_{Is}$ = 0.0131, and $K_{Ds}$ = 0.4400 at t = 9.5 s.

Figure 9 demonstrates the detailed speed response for the startup motor mode and switching to the sensorless control mode at the speed of 200 rpm. The ramp-up speed ratio is set at 500 rpm/s. As mentioned in step 1 of the *I-f* strategy, the rotor speed is increased, following the ramp function, with the initial reference current $i_q^*$ of 0.63 A. The actual rotor speed reaches 200 rpm in 0.415 s. At t = 0.441 s, the actual rotor speed reaches the maximum of 207.6 rpm, so the maximum error is +7.6 rpm. In step 2 of the *I-f* strategy, from t = 0.5 s, the reference current $i_q^*$ decreases down to 0.183 A, with a current down ratio of 0.42 A/s. The actual rotor speed is kept close to 200 rpm. At t = 1.585 s, the actual rotor speed is reduced to a minimum of 191.3 rpm, so the minimum error is −9.7 rpm. At t = 1.682 s, the switching operation occurs, when the deviation of the estimated position and the auto-generated position is 3.6 degrees. Three rotor position curves are almost overlapped together. The current $i_q$ has a small ripple, while the small pulse (about 0.05 A) appears in the current $i_d$. Therefore, the switching transient is smooth. Then, the motor begins to operate in the sensorless control mode.

Figure 10 shows the motor's speed responses while slowing down the rotor speed from 2000 to 400 rpm. The reference speed is sequentially varied as 2000 → 1700 → 1400 → 1200 → 800 → 1000 → 1200 → 1400 → 1000→ 700 → 400 rpm. Figure 10a shows that the estimated rotor speed is closely tracked by the reference speed and overlaps the actual rotor speed. Figure 10b presents the current response in the *d-q* axis. The current $i_q$ is also varied to the rotor speed. When the rotor speed is decreased, the external load becomes lower. Therefore, the less torque is required to the motor, and the current $i_q$ is regulated to be smaller. Moreover, the self-tuning PID controller's parameters are also tuned successfully. At t = 12.5 s (as $\omega_r$ = 1200 rpm), the PID gains are tuned to $K_{Ps}$ = 0.8490, $K_{Is}$ = 0.0115, $K_{Ds}$ = 0.4666. At t = 17.5 s (as $\omega_r$ = 1000 rpm), the PID gains are tuned to $K_{Ps}$ = 0.6567, $K_{Is}$ = 0.0094, $K_{Ds}$ = 0.4947. Lastly, the rotor speed decreases to 400 rpm and the PID gains are also tuned to $K_{Ps}$ = 0.6020, $K_{Is}$ = 0.0087, $K_{Ds}$ = 0.5026 at t = 19.5 s.

Figure 11 presents the system performance when changing the external load. In the beginning, the dynamic load system is operated with a capacitor of 470 µF, and the total resistor of 100 Ω, connected in the electrical circuit. The motor starts up and increases the rotor speed following the sequence of 200 → 500 → 1000 → 1500 → 2000 rpm. The rotor speed reaches 2000 rpm after 5 s. The current $i_q$ is approximately 1.79 A. At t = 6 s, the resistance load is changed from 100 Ω to 50 Ω by turning on some resistors in parallel connection. The external load is enhanced. The rotor speed reduces to the minimum of 1838 rpm, at t = 6.037 s, and stabilizes at 2000 rpm again at t = 6.468 s. The speed reduction is 162 rpm, and the recovery time is 0.431 s. The current $i_q$ is increased to 2.51 A. Additionally, at t = 8 s, the resistance load is returned to the initial value of 100 Ω by turning off the same resistors.

The motor speed is increased to the maximum of 2170 rpm at t = 8.041 s and stabilized at 2000 rpm again at t = 8.492 s. The speed increment is 170 rpm, and the recovery time is 0.451 s. The current $i_q$ is decreased to the previous value of 1.79 A because the additional external load is removed. It can be inferred that the motor still operates stably with the sensorless control algorithm by the novel SMO-PLL estimator in the dynamic load condition.

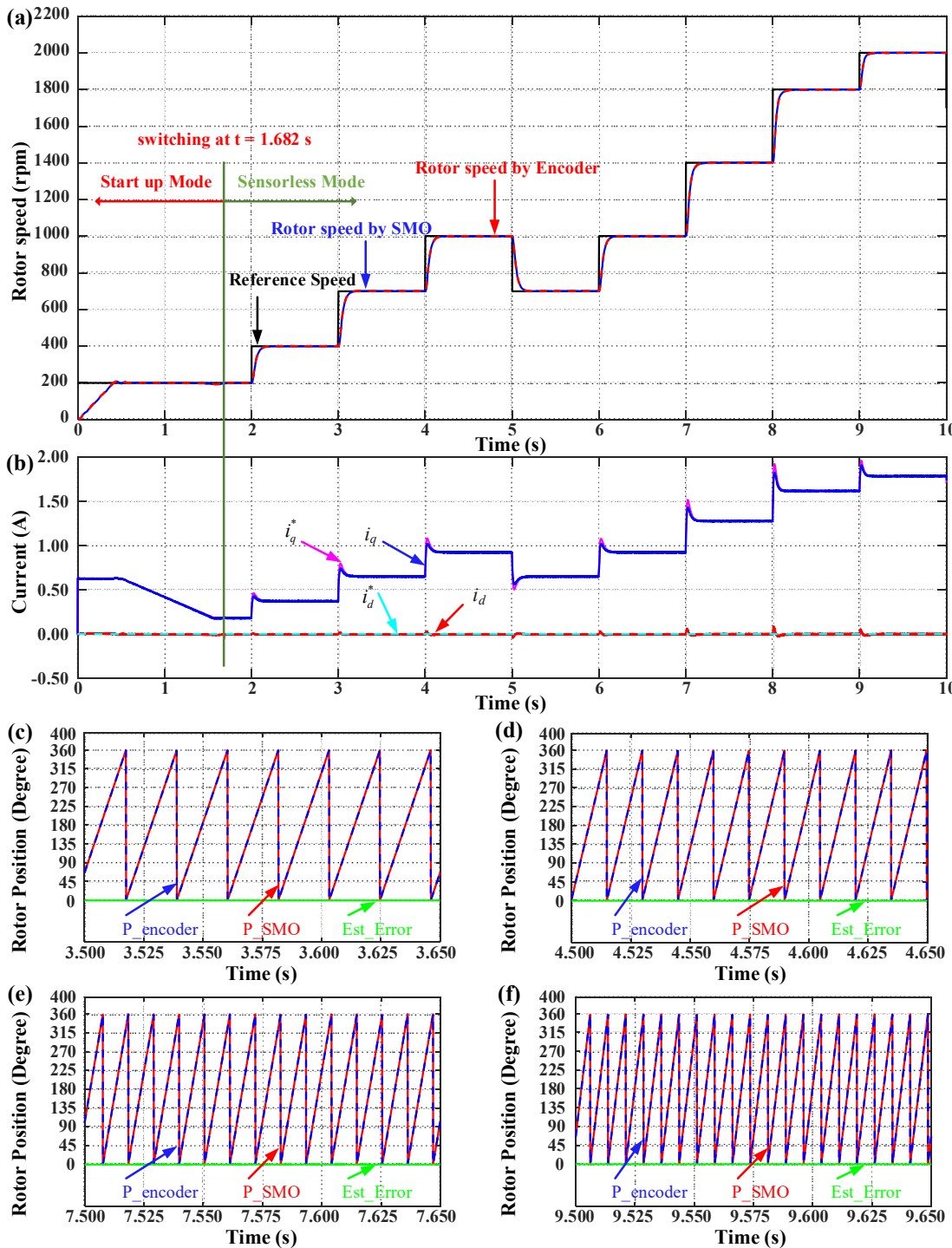

**Figure 8.** Simulation results in the speed increasing strategy, from 0 to 2000 rpm, for (**a**) speed response, (**b**) current response, and rotor position response at (**c**) 700, (**d**) 1000, (**e**) 1400, (**f**) 2000 rpm.

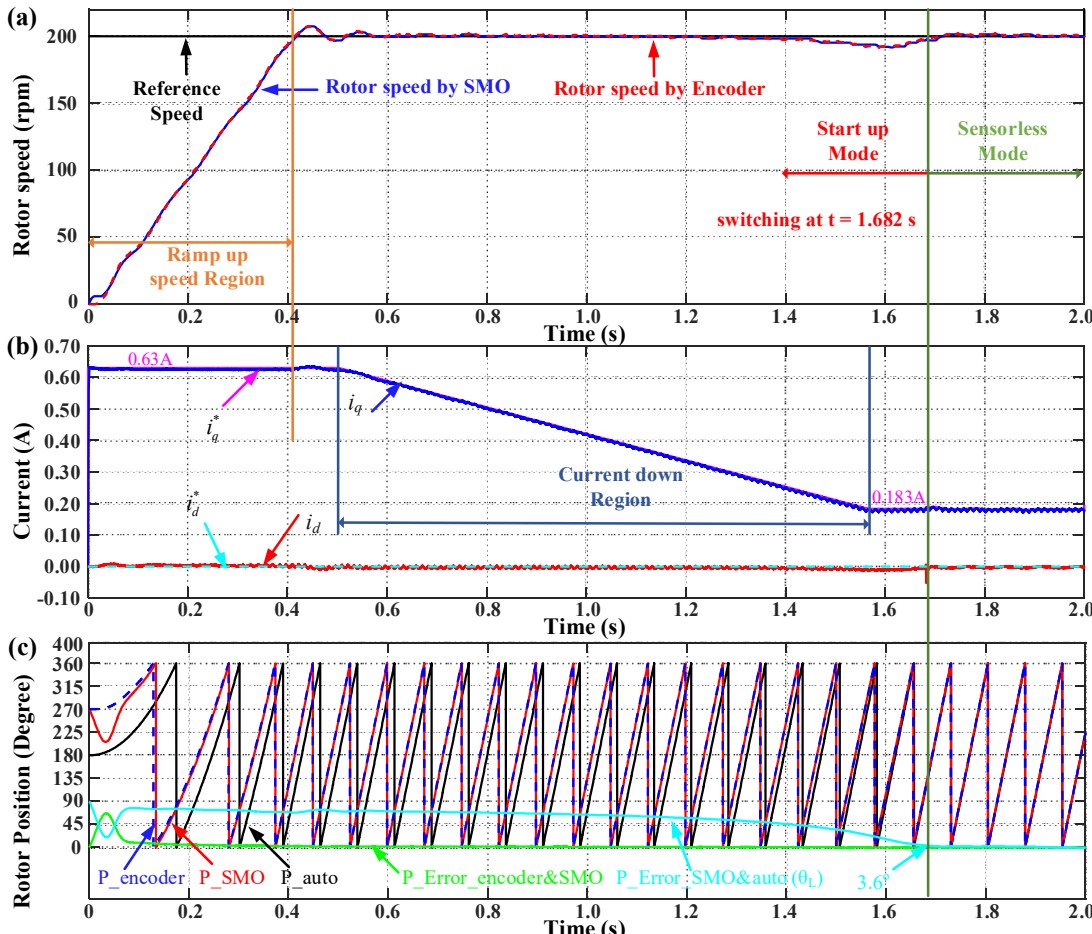

**Figure 9.** Simulation results in the startup mode and sensorless switching, from 0 rpm to 200 rpm for (**a**) speed response, (**b**) current response, (**c**) rotor position response.

As a summary, the various reference speed changes and the dynamic load condition are investigated in Figures 8–11 to analyze the controller performance. Referring to those results, it is easy to find that the rotor speed almost tracks to the commands very well, in which all the steady-state errors approach zero, and the overshoot or undershoot is also too small. The estimator works successfully because the estimated position approaches the actual position. The estimated error is almost zero. Moreover, the system has a good mode transition and robust performance against disturbance. The simulation results confirm that the proposed estimation and control algorithm for the sensorless PMSM drive system are correct and effective.

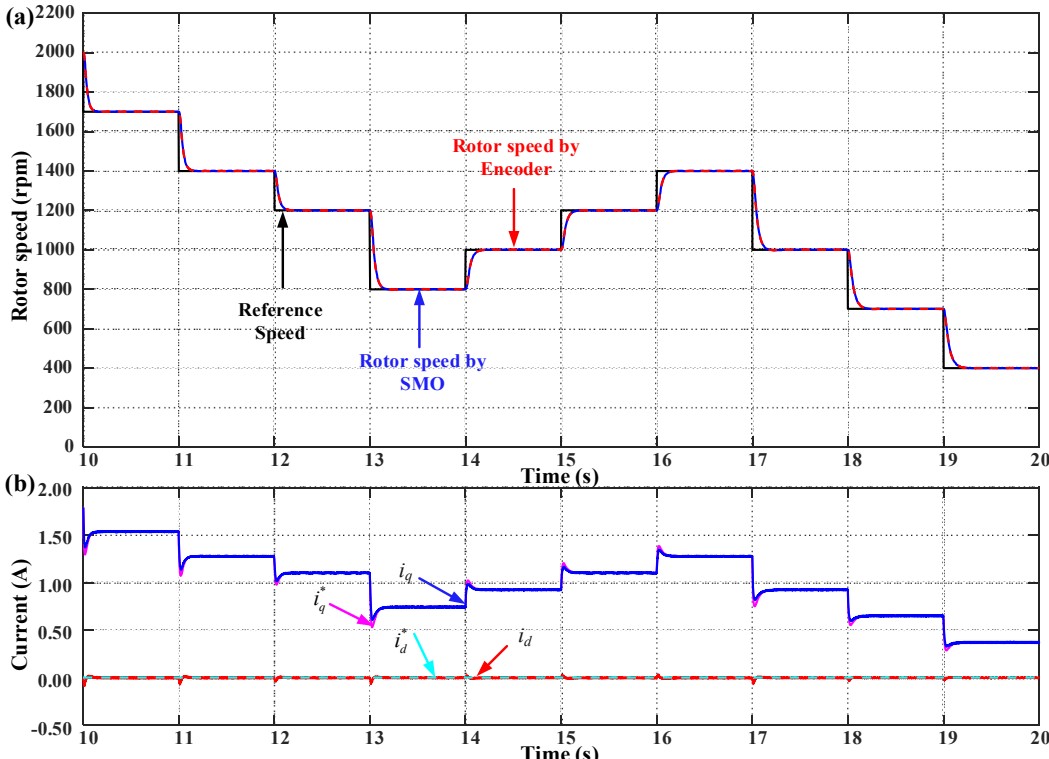

**Figure 10.** Simulation results in the speed decreasing strategy, from 2000 rpm to 400 rpm for (**a**) speed response, (**b**) current response.

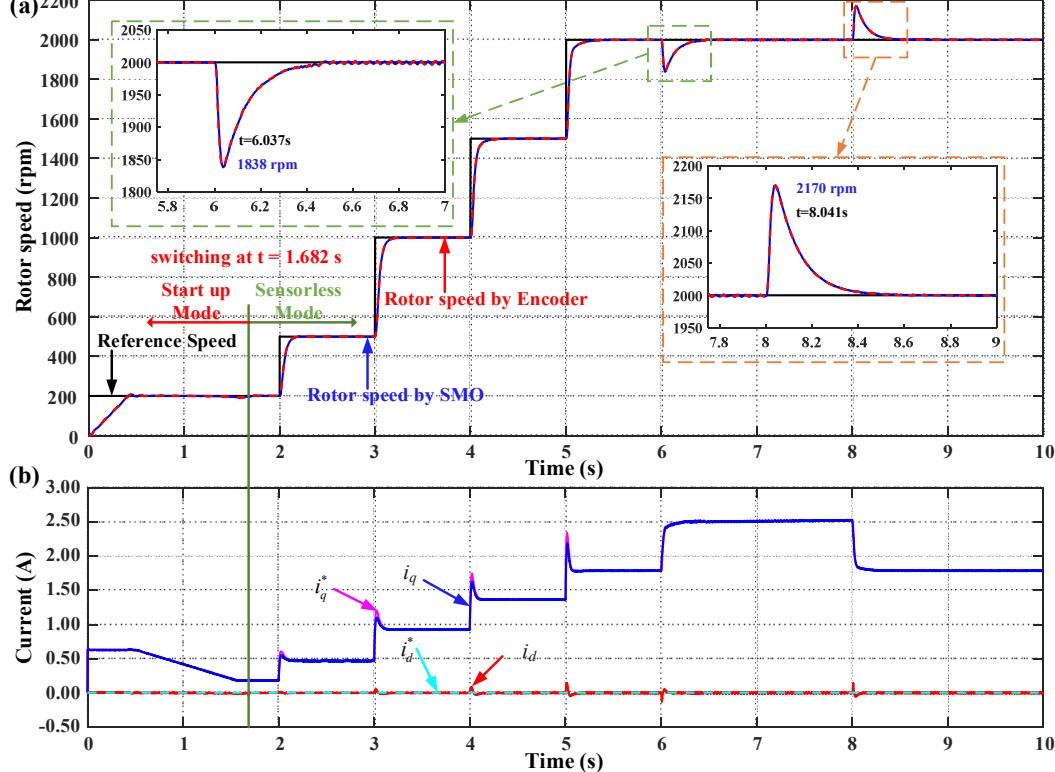

**Figure 11.** Simulation results in the case of the varied external load ($R_L = 100 \leftrightarrow 50\ \Omega$) at 2000 rpm for (**a**) speed response, (**b**) current response.

## 6. The Experimental Verification and Results

After finishing the simulation, to verify the algorithm and analyze the system performance, the motor control algorithm is implemented by the platform in Figure 12. The hardware platform consists of a PMSM coupled to a generator, with an electrical load system, an inverter, a DSP F28379D (as TI-DSP), and a control circuit. The parameters of PMSM are listed in Table 1. The DSP F28379D is equipped with 200 MHz dual C28xCPUs and dual CLAs, 1 MB Flash, 16-bit/12-bit ADCs, comparators, 12-bit DACs, HRPWMs, eCAPs, eQEPs, CANs, etc. The control circuit is designed to isolate the PWM signal between the inverter and TI-DSP, process the phase current measurement, and protect the overcurrent status. When the overcurrent happens, the PWM control signal generated by TI-DSP will be locked. The electrical load system includes a rectifier, a capacitor of 470 µF-450 V, and power resistors of 100 Ω-100 W.

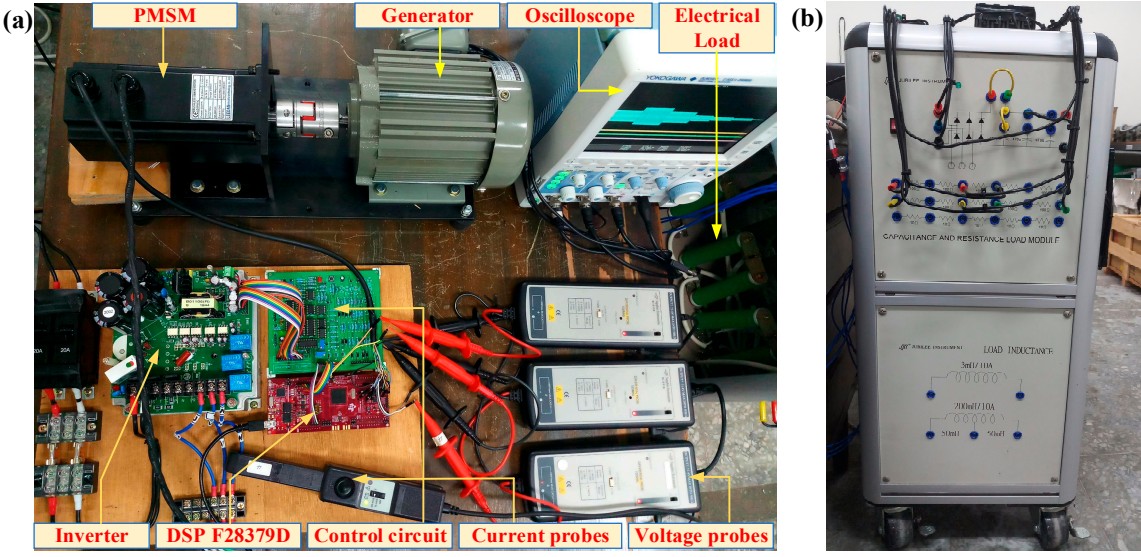

**Figure 12.** Experimental system with (**a**) real platform, (**b**) electrical load.

In the simulation structure in Figure 7, only the second block (B) is compiled to generate the algorithm code in C/C++ language, which is imported into the CCS software. The CCS software was responsible for connecting, downloading, debugging, and monitoring online the variables during the operation of the system. Moreover, some data in the block (C) are acquired online by MATLAB, based on the SCI function.

The sampling times of the velocity loop, the current loop and the novel SMO-PLL estimator, are still set as the same as the simulation values. The inverter's switching frequency is set at 15 Khz. The parameters of PI current controller are set as $K_{Pd} = 0.25$, $K_{Id} = 0.025$, $K_{Pq} = 0.25$, and $K_{Iq} = 0.025$. The RBFNN's initial parameters are setup as the node centers ($c_1 = -0.0025$, $c_2 = -0.00125$, $c_3 = 0.0$, $c_4 = 0.00125$, $c_5 = 0.0025$), the node widths ($b_1 = b_2 = b_3 = b_4 = b_5 = 0.5$), the connective weights ($w_1 = w_2 = w_3 = w_4 = w_5 = 0.0000625$). The learning rate is 0.435 and the momentum factor is 0.75. Additionally, the discrete PID's initial parameters are designed as $K_{Ps} = 0.735$, $K_{Is} = 0.00435$, and $K_{Ds} = 0.478$. All experimental results are implemented with an initial resistance load of 100Ω-400 W, a capacitor of 470µF-450V in the dynamic load system.

Figure 13 shows the motor's speed responses while speeding up the rotor speed from 0 to 2000 rpm. The reference speed is changed in a period of 5s with the sequences as 300 → 500 → 700 → 1000 → 800 → 1200 → 1600 → 1800 → 2000 rpm. Figure 13a refers that the novel SMO-PLL estimated rotor speed is closely tracked to the reference speed, and overlapped the actual rotor speed, measured by the encoder. Although there is the oscillation of the current responses in Figure 13b, their average value still follows the current command. Figure 13c–f illustrates the electrical positions of the rotor,

separately calculated by the novel SMO-PLL estimator and the encoder, with the estimated errors at the speeds of 500, 1000, 1600, 2000 rpm. There are approximately 5, 10, 16 and 20 position cycles in a period of 0.15s, respectively. Correspondingly, the rotation frequencies are 33.33, 66.67, 106.67, and 133.33 Hz. These values are suitable for the rotor speed. The estimated and actual positions are approximated to each other, so the estimated error approaches zero. The self-tuning PID controller's parameters are also updated during the operating time. The PID gains are tuned to $K_{Ps} = 0.7378$, $K_{Is} = 0.0039$, $K_{Ds} = 0.4751$ at t = 12 s (as $\omega_r$ = 500 rpm); $K_{Ps} = 0.7583$, $K_{Is} = 0.0039$, $K_{Ds} = 0.4692$ at t = 22 s (as $\omega_r$ = 1000 rpm); $K_{Ps} = 0.8408$, $K_{Is} = 0.0034$, $K_{Ds} = 0.4619$ at t = 37.5 s (as $\omega_r$ = 1600 rpm). Lastly, rotor speed reaches 2000 rpm, and $K_{Ps} = 0.8672$, $K_{Is} = 0.0034$, $K_{Ds} = 0.4487$ at t = 46 s.

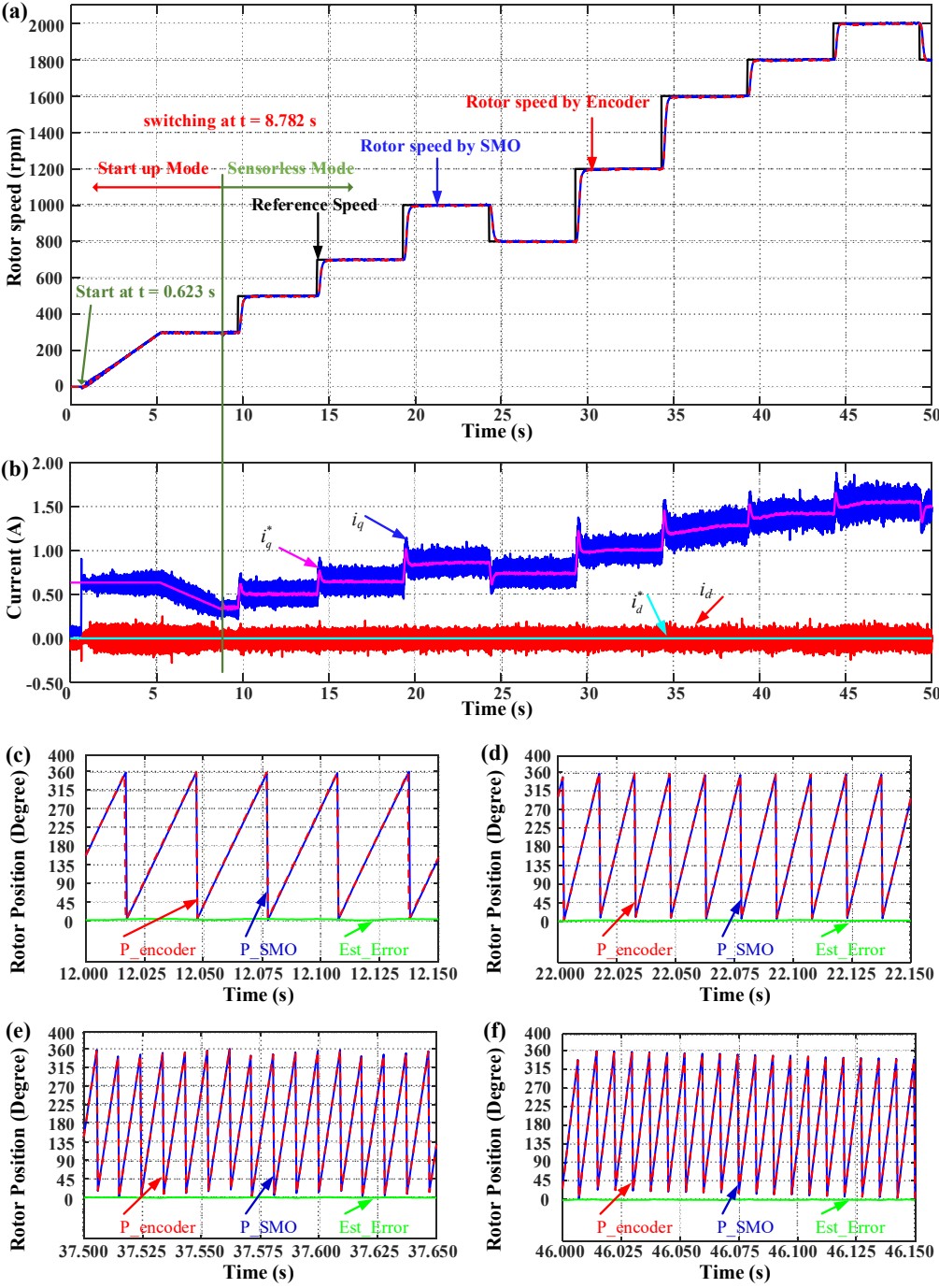

**Figure 13.** Experimental results in the speed increasing strategy, from 0 to 2000 rpm for (**a**) speed response, (**b**) current response, and rotor position response at (**c**) 500, (**d**) 1000, (**e**) 1600, (**f**) 2000 rpm.

Figure 14 demonstrates the detailed speed response for the startup motor mode and switching to the sensorless control mode at the speed of 300 rpm. The ramp-up speed ratio is set up at 66.67 rpm/s. At t = 0.623 s, the motor starts to speed up. The rotor speed is increased, following the ramp function with the initial reference current $i_q^*$ of 0.635 A. The actual rotor speed reaches 295 rpm at t = 5.2 s. Then, the reference current $i_q^*$ decreases down to 0.336 A with a current down ratio of 0.085 A/s. The reference speed is kept at 300 rpm, and the actual rotor speed is still close to 300 rpm. At t = 6.403 s, the actual rotor speed reaches the maximum of 302.7 rpm. The maximum speed error is +2.7 rpm. At t = 8.832 s, the actual rotor speed drops to a minimum of 286.1 rpm. The minimum speed error is −13.9 rpm. The estimated position leads the auto position by 27.95 degrees at t = 5 s, while the estimated position error is 39.02 degrees. These values are reduced in the current down region. The switching operation occurs at t = 8.782 s when the deviation of estimated position and auto position ($\theta_L$) is also equal to 3.6 degrees. The estimated position error is only 0.175 degrees. The ripple of the current $i_q$ is roughly 0.06 A, while the current $i_d$ fluctuates around zero within a range of 0.1 A. The motor switches the control mode smoothly. After this, the motor begins to operate in the sensorless control mode.

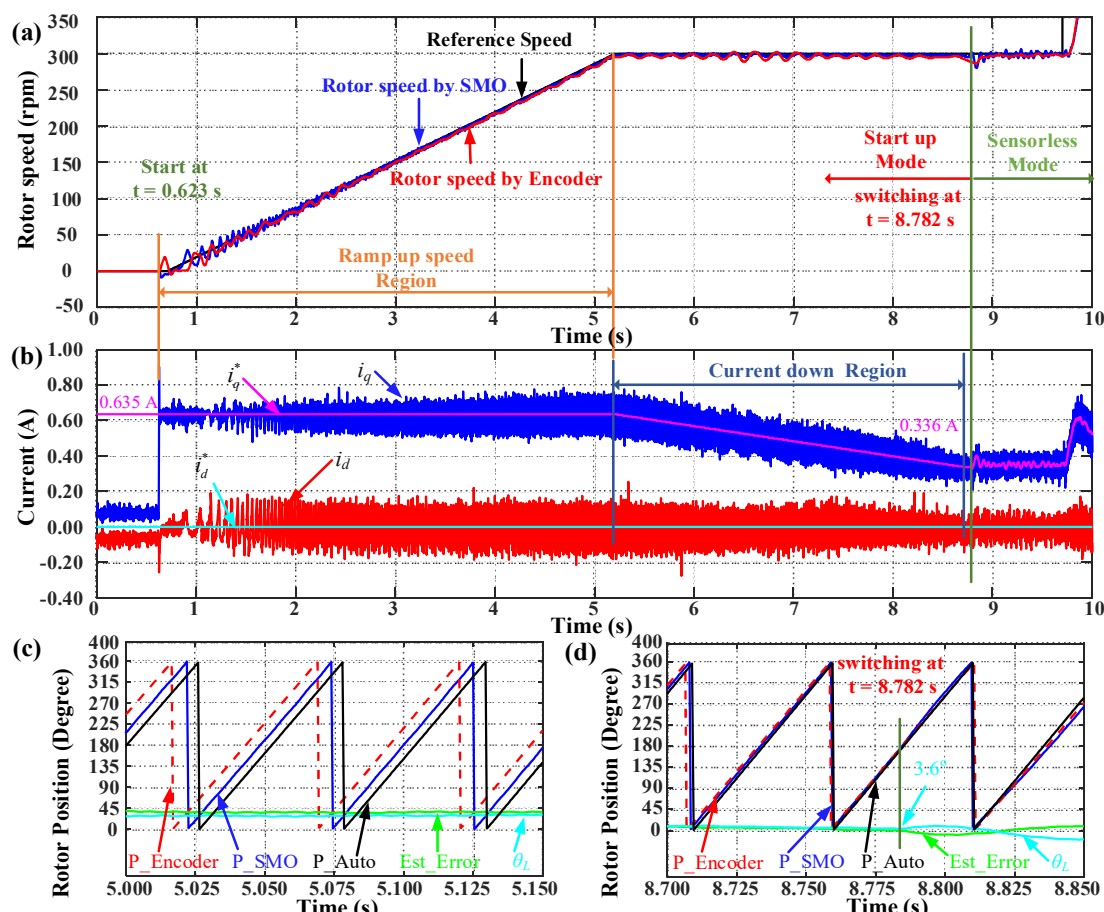

**Figure 14.** Experimental results in the startup mode and sensorless switching, from 0 rpm to 300 rpm, for (**a**) speed response, (**b**) current response, (**c**) rotor position response.

Figure 15 shows the motor's speed responses while slowing down the rotor speed from 1800 to 300 rpm. The reference speed is sequentially varied as 1800 → 1600 → 1400 → 1000 → 1200 → 800 → 500 → 300 → 500 → 700 → 1000 rpm. Figure 15a shows that the estimated rotor speed is closely tracked by the reference speed and overlaps the actual rotor speed. Figure 15b presents the current response in the *d–q* axis. The current $i_q$ is also varied according to the rotor speed. When the rotor speed is decreased, the external load becomes lower. Therefore, the less torque is required by the motor, and the current $i_q$ is regulated to be smaller. Moreover, the self-tuning PID controller's

parameters are also tuned effectively. The PID gains are tuned to $K_{Ps}$ = 0.832, $K_{Is}$ = 0.0039, $K_{Ds}$ = 0.4531 at t = 62 s (as $\omega_r$ = 1400 rpm); $K_{Ps}$ = 0.7280, $K_{Is}$ = 0.0044, $K_{Ds}$ = 0.4580 at t = 77 s (as $\omega_r$ = 800 rpm); and $K_{Ps}$ = 0.7046, $K_{Is}$ = 0.0044, $K_{Ds}$ = 0.4639 at t = 87 s (as $\omega_r$ = 300 rpm).

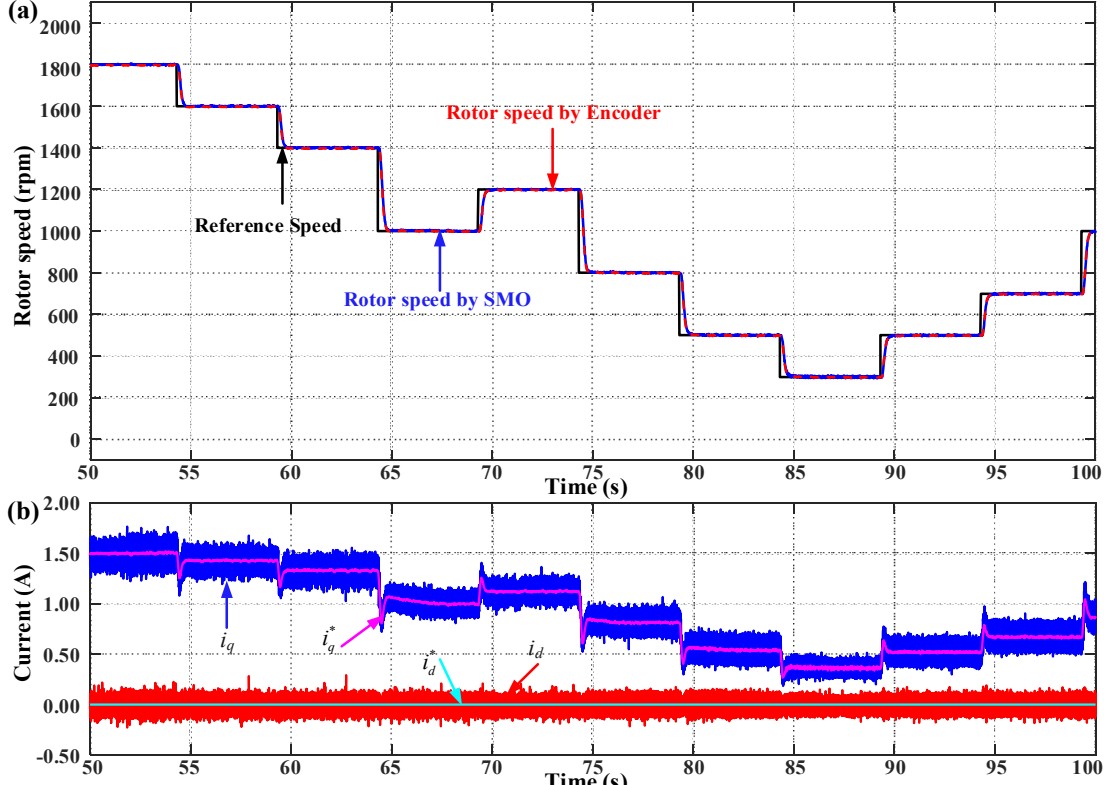

**Figure 15.** Experimental results in the speed decreasing strategy, from 1800 rpm to 300 rpm for (**a**) speed response, (**b**) current response.

Figure 16 presents the system performance when the external load is varied. In the beginning, the dynamic load is set up with a capacitor of 470 μF-450 V, the total resistor of 100 Ω-400 W. The motor starts up and increases the rotor speed, reaching 2000 rpm. The current $i_q$ fluctuates around 1.56 A. At t = 37.99 s, the total resistance load is changed to 50 Ω-800 W by turning on more resistors. At t = 38.11 s, the actual rotor speed (red line) drops to a minimum of 1797 rpm while the minimum estimated value (blue line) is 1817 rpm. The recovery time is 2.99 s when the motor reaches 1995 rpm again at t = 40.98 s. The average current $i_q$ reaches 2.42 A due to the external load increment. In addition, at t = 45.60 s, the total resistance load is returned to the initial value of 100 Ω-400 W by turning off the same resistor. The actual rotor speed is increased to the maximum of 2171 rpm at t = 45.71 s, while the maximum estimated value is 2157 rpm. The recovery time is 1.06 s when the motor stabilizes at 2000 rpm again at t = 46.66 s. The average current $i_q$ is decreased to 1.57 A because of the external load reduction. Within the speed regulation at 2000 rpm, the estimate speed error is about +5 rpm, while this error is larger at the up peak (20 rpm) or the low peak (14 rpm). The maximum control speed steady-state error is ±5 rpm at 2000 rpm. Although the system is affected by the disturbance of the external load, the motor still operates stably and successfully with the sensorless control algorithm based on the novel SMO-PLL estimator.

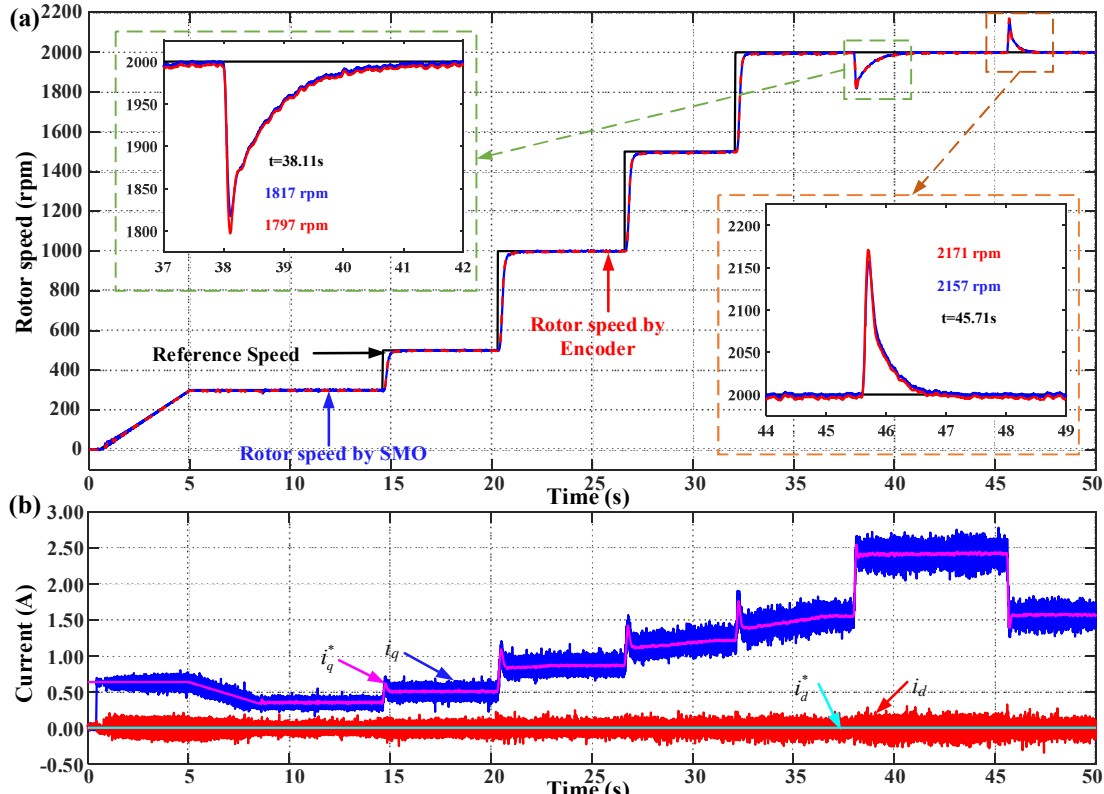

**Figure 16.** Experimental results in the case of the varied external load (RL = 100 ↔ 50 Ω) at 2000 rpm for (**a**) speed response, (**b**) current response.

Finally, similar to the simulation, the experimental results in Figures 13–16 show that the rotor speed almost tracks to the command very well, all the steady-state errors approach zero (within ±5 rpm in tolerance), and the overshoot or undershoot is also too small. The novel SMO-PLL estimator works successfully. The estimated position approaches the actual position. The estimated error is close to zero. Additionally, the real-time system has a good mode transition and robust performance against disturbance. The experimental results again confirm that the proposed estimation and control algorithm of the PMSM system are correct and effective in the real-time system. Furthermore, the DSP application for the PMSM drive control system is built in MATLAB Simulink properly, and is deployed to CCS software to realize the real-time system successfully. This deployment method shortens the application development time.

## 7. Conclusions

In this paper, a self-tuning PID controller based on a radial basis function neural network and a rotor position estimator based on the novel SMO in combination with a PLL for the sensorless PMSM drive control system have been described and developed successfully. The control algorithm, consisting of the *I-f* startup strategy, novel SMO-PLL estimator, RBFNN-based self-tuning PID controller, and FOC algorithm, was deployed properly in both the simulation and the real-time hardware, established on a DSP F28379D. The system performance has been verified in three terms: startup mode, tracking response, and speed regulation with the dynamic load system. The simulation and experimental results indicate that the motor control system has a smooth transition from the startup mode to the sensorless control mode with the low ripple in the current and the rotor speed. The novel SMO-PLL estimator is stable, and the estimated position approximates to the actual position, so the estimated error is almost minimal and negligible. The PID gains are tuned effectively. The rotor speed tracks properly to the reference speed, and the overshoot or undershoot is very small. The steady-state

errors approach zero. Additionally, the system provides a robust performance against disturbance. Accordingly, the system performance confirms that the proposed intelligent control algorithm and the position estimator for the sensorless PMSM drive control system are correct and effective. Furthermore, the DSP application for the PMSM drive control system is implemented based on the combination of MATLAB and CCS software. Therefore, we can easily develop additional intelligent controllers for the motor control system. It also has the advantage of faster testing, monitoring and acquisition of data online, and troubleshooting of systems. This motivates the improvement of the proposed algorithm for the wide speed range control in further works.

**Author Contributions:** H-K.H. wrote this article, designed the control method, implemented the hardware platform, drew the figures and performed the simulations as well as the experimental data. S-C.C. supervised, coordinated the investigations, checked up the manuscript's logical structure. H.T. was responsible for exchanging ideas, reviewing the article draft. All authors have read and agreed to the published version of the manuscript.

**Funding:** This research received no external funding.

**Conflicts of Interest:** The authors declare no conflict of interest.

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
