# Peer review of "Realization of the Sensorless Permanent Magnet Synchronous Motor Drive Control System with an Intelligent Controller"

_electronics, doi:10.3390/electronics9020365_

Round 1

Reviewer 1 Report

In my opinion, the paper shows poorly presented equations, with Chinese symbols on top of several equations throughout the document. A lot of them.

The title is too long and with acronyms not allowing us to understand what the work is when we read the paper title. In titles, acronyms that have not yet been defined should not be used.

Some figures are too small and do not allow full visibility and understanding. Also the figures and results explanation should be improved.

Reviewer 2 Report

The authors present an interesting method, based on Neural Networks, for controlling a Permanent Magnet Synchronous Motor through a sensorless control algorithm. Regarding this paper I would like to make the following remarks.

I consider being fundamental for the authors to revise the multiple Chinese characters throughout the paper. It prevents the equations to be easily understood.

The authors should also define every abbreviation in their first use; e.g. PID in 58.

Furthermore, abbreviations should only be defined once; e.g. DSP in lines 32 and 64.

Finally, Conclusions section should include a deeper interpretation of the results, instead of rely solely on summarizing the achievements.

Round 2

Reviewer 1 Report

Authors improved the paper according reviewers suggestions.

Please remove the acronyms definitions from the keywords. Use only words as  keywords.